# Enhanced light-matter interaction in an atomically thin semiconductor coupled with dielectric nano-antennas

L. Sortino [1]*, P.G. Zotev[1], S. Mignuzzi[2], J. Cambiasso[2], D. Schmidt[3], A. Genco[1], M. Aßmann [3], M. Bayer [3], S.A. Maier[2,4], R. Sapienza [2] & A.I. Tartakovskii [1]*

Unique structural and optical properties of atomically thin two-dimensional semiconducting transition metal dichalcogenides enable in principle their efficient coupling to photonic cavities having the optical mode volume close to or below the diffraction limit. Recently, it has become possible to make all-dielectric nano-cavities with reduced mode volumes and negligible non-radiative losses. Here, we realise low-loss high-refractive-index dielectric gallium phosphide (GaP) nano-antennas with small mode volumes coupled to atomic mono- and bilayers of $WSe_2$. We observe a photoluminescence enhancement exceeding $10^4$ compared with $WSe_2$ placed on planar GaP, and trace its origin to a combination of enhancement of the spontaneous emission rate, favourable modification of the photoluminescence directionality and enhanced optical excitation efficiency. A further effect of the coupling is observed in the photoluminescence polarisation dependence and in the Raman scattering signal enhancement exceeding $10^3$. Our findings reveal dielectric nano-antennas as a promising platform for engineering light-matter coupling in two-dimensional semiconductors.

[1] Department of Physics and Astronomy, University of Sheffield, Sheffield S3 7RH, UK. [2] The Blackett Laboratory, Department of Physics, Imperial College London, London SW7 2BW, UK. [3] Experimentelle Physik 2, Technische Universität Dortmund, 44221 Dortmund, Germany. [4] Chair in Hybrid Nanosystems, Nanoinstitute Munich, Faculty of Physics, Ludwig-Maximilians-Universität München, 80539 Munich, Germany. *email: l.sortino@sheffield.ac.uk; a.tartakovskii@sheffield.ac.uk

**M**onolayer semiconducting transition metal dichalcogenides (TMDs)[1] such as WSe$_2$ exhibit bright excitonic luminescence and strong absorption at room temperature with potential for photonic applications[2,3]. An important property favouring integration in devices is their compatibility with a wide range of substrates. So far, photonic device demonstrations include monolayer TMDs coupled to nano-cavities in photonic crystals[4,5], nanobeam waveguides[6], and to microdisk cavities[7]. TMD monolayers and van der Waals heterostructures[8] comprised of vertically stacked atomic layers of TMDs, hexagonal boron nitride and graphene have been integrated in Fabry–Perot microcavities[10,11]. These devices provide photonic modes with relatively high-quality factors, $Q$, in the range of 100s to 1000s. Despite the coupling to TMD monolayers via the relatively weak evanescent field, lasing in hybrid TMD dielectric cavities has been observed[4,6,7]. Moreover, the strong light–matter interaction regime has been realised in optical microcavities[9–12] and photonic crystals[5], where atomic layers of two-dimensional (2D) TMDs were placed at the anti-node of the photonic mode. Most of these devices relied on confining electromagnetic fields in diffraction-limited volumes, $V_{eff} \gtrsim (\lambda/n)^3$, in order to increase the spontaneous emission rate by the Purcell enhancement factor $F_p$, which scales as $Q/V_{eff}$. Whereas the high $Q$ has been readily realised, $V_{eff}$ provided by these structures is relatively large, leading to modest values of $F_p$. Most of these devices also show a reduction of the light intensity compared with bare TMD monolayers, explained by the presence of fast non-radiative processes in the currently available TMDs, where the quantum yield is typically <0.1%[13].

The effective volume of the optical mode can be reduced below the diffraction limit in plasmonic nano-cavities and nano-antennas[14,15]. By coupling semiconducting TMDs to such plasmonic structures, large photoluminescence (PL) enhancements[16–21], strong light–matter coupling[22–24], brightening of the dark excitonic states[25], and modification of optical properties of quantum light emitters[26,27] have been observed. In some of these reports, special care had to be taken to overcome optical losses in metallic plasmonic structures by introducing a few nm dielectric spacer separating the TMD layer[20,21,29]. This turns out to be particularly important to suppress quenching for quantum light emitters[26,27].

Recently, it has also been shown that high-refractive-index dielectric nano-antennas can provide confined optical modes with strongly reduced mode volumes[28,30–34]. The main advantages of such structures are low non-radiative losses induced in the coupled light-emitting material[28,30–32]. Experimentally, strong fluorescence enhancement and radiative lifetime shortening by a factor >20 has been shown for dye molecules coupled to GaP cylindrical nano-antennas (refractive index $n > 3$)[32]. Furthermore, it has been shown that such nano-antennas can be designed in principle to provide Purcell enhancements of thousands[31]. On the other hand, recently, modified directionality of PL was shown for monolayer MoS$_2$ coupled to a Si nanowire[33]. Si nano-particles coupled to WS$_2$ were explored for possibilities to realise the strong light–matter interaction[34]. Finally, multilayer TMDs themselves were used to fabricate high-index nanodisks, whose resonant response could be tuned over the visible and near-infrared (near-IR) ranges[35].

Here we report large enhancements of the PL and Raman scattering intensity in monolayer (1L) and bilayer (2L) WSe$_2$ placed on top of cylindrical GaP nano-antennas (Fig. 1a), compared with WSe$_2$ on the planar GaP. The incident radiation interacts with the nano-antennas, and excites a mode strongly localised around the pillar edges. This confinement effect is observed in a broad spectral range, overlapping with the optical response of both monolayer and bilayer WSe$_2$. Our approach exploits the extreme ability of the atomically thin layers of TMDs to stretch and conform to the nano-structured surfaces and therefore favourably align themselves with the confined optical mode. Our interpretation of the observed PL enhancement shows that it arises from a combination of the Purcell enhancement, efficient redirection of the emitted PL in the space above the substrate, and larger absorption of light in the 2D layer related to the enhancement of the optical field in the confined mode. Similarly to experiment, our model shows an increase of the PL enhancement as the pillar radius reduces. Supporting our interpretation, time-resolved measurements show the shortening of the PL lifetime, which goes beyond our time resolution, only enabling a lower-bound estimate of the PL decay rate enhancement by a factor of 6. We further confirm the photonic effect of the nano-antennas by demonstrating linearly polarised PL and enhanced Raman scattering in the coupled WSe$_2$. Our findings show an effective approach to engineering light–matter coupling at the nanoscale, by exploiting the low-loss optical modes of the dielectric nano-antennas together with the unique mechanical and optical properties of 2D semiconductors.

## Results

**Coupling WSe$_2$ atomic layers with dielectric nano-antennas.** The calculated spatial distribution of the electric field around the cylindrical double-pillar GaP nano-antennas (referred to as dimers below) is shown in Fig. 1b, c. Light at a pumping wavelength of 685 nm polarised linearly along ($X$) and perpendicular ($Y$) to the line connecting the centres of the pillars is used. $(|E|/|E_0|)^2$ is shown for the plane containing the top surface of the dimer, i.e. 200 nm above the planar substrate. Here $E$ and $E_0$ are the electric field amplitudes of the wave scattered by the pillars and the normally incident wave, respectively. The 2D distribution in the plane is shown as a colour map. In the small volume within the dimer gap, an enhancement is only observed for the $X$-polarisation[32,37] (see further details in Supplementary Note I). The graphs shown with white lines present the variation of $(|E|/|E_0|)^2$ along the horizontal and vertical dotted lines, revealing strong maxima at the edges of the pillars. For the $X$-polarisation, at this height of 200 nm, these maxima are stronger than the $(|E|/|E_0|)^2$ values in the gap. They are also stronger than the enhancement values at the pillar edges in the $Y$-polarisation.

Atomic force microscopy (AFM, Fig. 1d) shows that the transferred atomically thin layer of WSe$_2$ closely follows the shape of the dimer, thus strongly overlapping with the volume of the confined optical mode. As shown in Fig. 1e, the spectral response of the nano-antenna is very broad, extending well into the visible and near-IR ranges, and fully overlapping with the PL spectra of both 1L- and 2L-WSe$_2$ (see also Supplementary Note II).

Figure 1f shows a PL image of 1L and 2L samples deposited on an array of nano-antennas and measured using an optical microscope. The image is recorded using the techniques of ref. [38] with unpolarised white light illuminating the sample through a short-pass filter and a long-pass filter installed in the imaging path. Bright PL replicating the shape of the dimers is visible for both 1L (yellow) and 2L (purple), whereas the PL from WSe$_2$ on planar GaP is very weak (dark areas around the pillars). A comparison of the intensities in the PL and dark field microscopic images is shown in Fig. 1g, where the intensities are measured along the dotted line shown for one of the nano-antennas in Fig. 1f. The PL enhancement is observed most strongly around the edges and in the gap of the nano-antenna, where, as seen in the dark field profile, most of the light is scattered by the pillars. Further comparison of the PL, bright-, and dark-field images is given in Supplementary Note III.

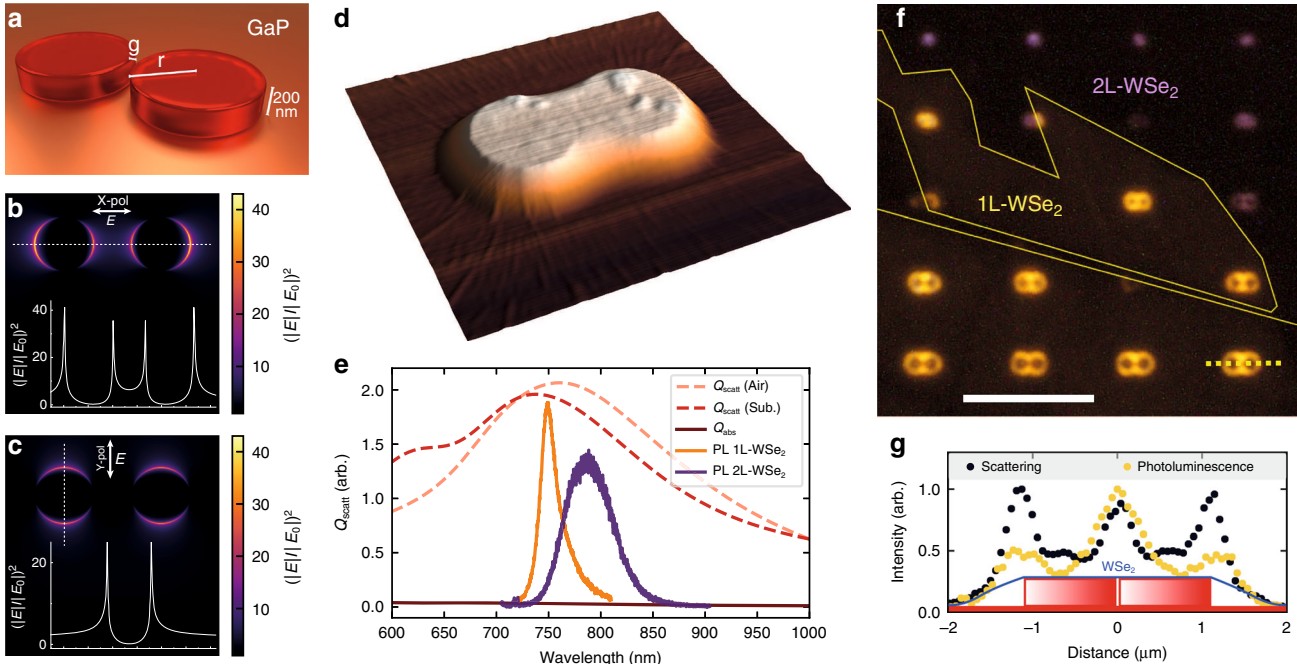

**Fig. 1** Coupling of an atomically thin semiconductor to a dielectric nano-antenna. **a** Schematic view of a GaP dimer nano-antenna with a height $h = 200$ nm, gap width $g$, and pillar radius $r$. **b**, **c** Calculated relative intensity $(|E|/|E_0|)^2$ of light at 685 nm scattered by a GaP dimer with $r = 50$ nm, $h = 200$ nm, and $g = 65$ nm. $E$ ($E_0$) is the electric field amplitude of the light scattered by (normally incident on) the dimer. The polarisation of the incident light is shown with arrows. $(|E|/|E_0|)^2$ is calculated at the height of 200 nm corresponding to the top surface of the pillars. The inset shows the variation of $(|E|/|E_0|)^2$ along the horizontal and vertical dotted lines. **d** Atomic force microscopic (AFM) image of a GaP nano-antenna ($r = 500$ nm) covered with a monolayer $WSe_2$. **e** Simulated scattering ($Q_{scatt}$) and absorption ($Q_{abs}$) efficiency integrated over numerical aperture NA = 0.9 for a GaP nano-antenna ($g = 35$ nm, $r = 100$ nm) illuminated with a plane wave. Pink (red) dashed line $Q_{scatt}$ for light scattered upwards in air (downwards into the substrate). Room temperature PL spectra are shown for 1L-$WSe_2$ (orange) and 2L-$WSe_2$ (purple). **f** Optical microscopic image showing PL from dimer nano-antennas array covered with 1L- and 2L-$WSe_2$ illuminated with unpolarised white light (see 'Methods' for details). Scale bar is 10 μm. The yellow lines show the boundaries between 1L- and 2L-$WSe_2$. PL from 1L (2L) sample shows as false yellow (purple) in the image. **g** Intensity profiles extracted from a dark-field (black) and PL (yellow) microscopic images for a dimer with $r = 500$ nm measured along the dashed line in **e**. The intensity profiles are overlaid with the schematic of a dimer

**PL enhancement factor**. We have additionally carried out detailed room temperature PL measurements in our micro-PL set-up for 1L- and 2L-$WSe_2$ placed on GaP nano-antennas. We use a laser with wavelength $\lambda = 685$ nm, which is below the GaP absorption edge and is absorbed only in the $WSe_2$ layer. Figure 2a, b show PL spectra for 1L- and 2L-$WSe_2$ coupled to GaP nano-antennas with $r = 300$ nm and $r = 100$ nm, respectively, and compare them with PL from the 2D layers placed on the planar GaP. Strong enhancement of PL intensity exceeding 50 times for $WSe_2$ placed on nano-antennas is observed. Lower PL intensity for 2L-$WSe_2$ is a consequence of its indirect bandgap, in contrast to the 1L-$WSe_2$ having a direct bandgap. The effect of strain present in $WSe_2$ placed on the nano-pillar is evident in the PL redshift for 1L and spectral modification for 2L samples.

We compare the observed PL intensity for $WSe_2$ coupled to the nano-antennas ($I_{on}$) to that of the uncoupled $WSe_2$ on planar GaP ($I_{off}$) by introducing the PL enhancement factor[15,19,39,40], $\langle EF \rangle$, defined as:

$$\langle EF \rangle = \frac{I_{on}}{A_{dimer}} \left( \frac{I_{off}}{A_{laser}} \right)^{-1} \quad (1)$$

Here we take into account the considerable difference between the PL collection area $A_{laser}$ defined by the excitation laser spot of 3.5 μm diameter, and the geometrical area for a given dimer, $A_{dimer} = 2 \times \pi r^2$. We expect that $A_{dimer}$ is larger than the actual area from where the enhanced PL is collected (mostly the edges of the pillars and the dimer gap). Thus, $\langle EF \rangle$ calculated in this way is expected to be a lower-bound estimate of the observed effect.

Figure 2c shows the $\langle EF \rangle$ values extracted from the experimental data for different nano-antennas. $\langle EF \rangle$ for 1L-$WSe_2$ exhibits an increase from $10^2$ for the large $r = 500$ nm pillars to nearly $10^4$ for $r = 50$ nm, whereas the variation is more pronounced for the 2L samples, where $\langle EF \rangle$ changes from $\approx 30$ to $4 \cdot 10^4$. Such large $\langle EF \rangle$ values are comparable with the highest reported in plasmonic/TMDs systems[19].

As we show below, the observed enhancement is the consequence of the interaction of $WSe_2$ with the optical mode of the nano-antenna. The extremely efficient overlap between the 2D layers and the optical mode field maxima are important for enhancing the interaction with the nano-antennas. Figure 2c shows that there is a variation of $\langle EF \rangle$ between the antennas of the same size. There are several factors that can cause this. (1) Non-uniformity of the coupling between $WSe_2$ and the nano-antennas caused by a variety of factors such as local contamination from the polymer used for the $WSe_2$ transfer, local deformation of $WSe_2$, local presence of water, etc. (2) Non-uniformity of the structural properties of the nano-antennas. For example, the size of the gap may vary. The quality of etching may also vary, for example, producing side walls of the pillars, which are not perfectly vertical, etc.

A smaller value of $\langle EF \rangle$ for the 2L sample for the nano-antennas with the large radii is probably due to its higher rigidity compared with 1L. As the radius becomes smaller, the 2L conforms more closely with the shape of the nano-antenna, and in addition, the increased strain in the crystal may lead to the indirect to direct bandgap crossover[41], yielding larger values of $\langle EF \rangle$.

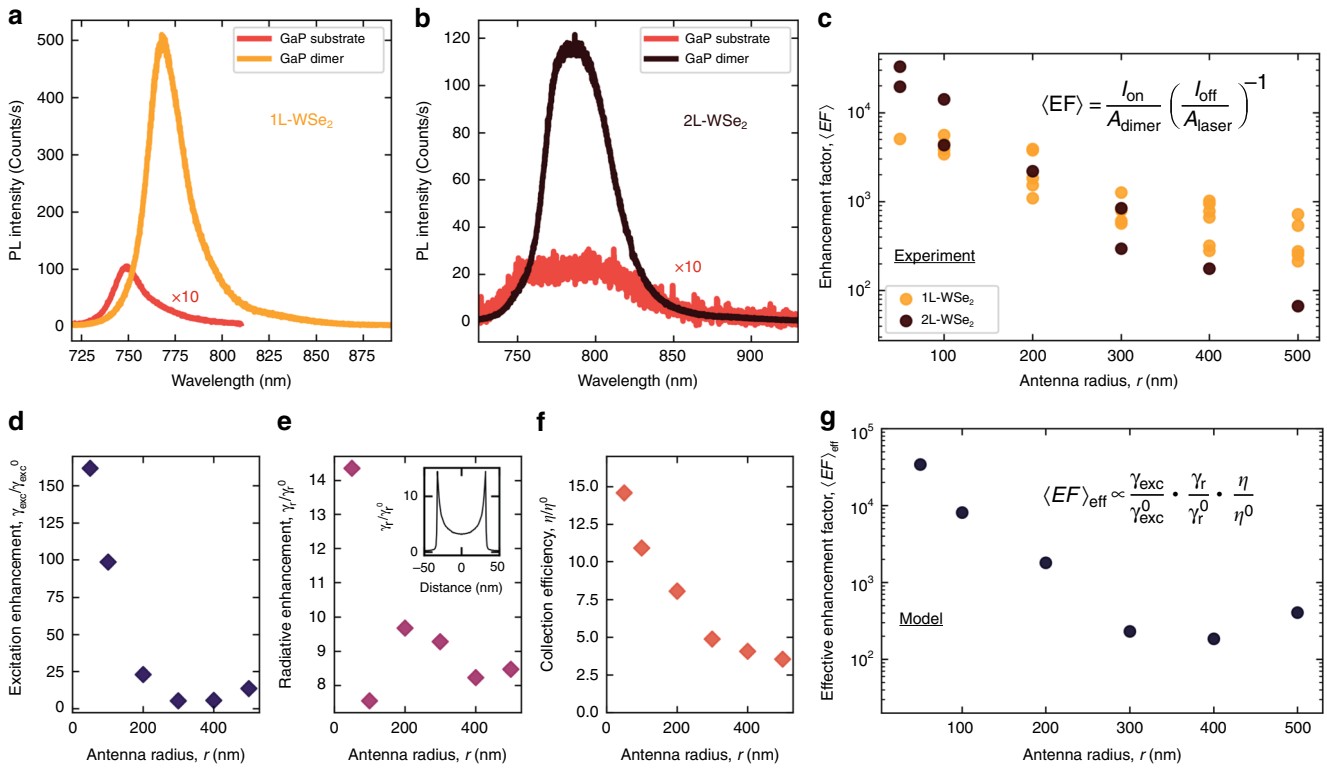

**Fig. 2** Photoluminescence enhancement of monolayer and bilayer WSe$_2$ coupled to GaP nano-antennas. **a**, **b** PL spectra for 1L-WSe$_2$ (orange) and 2L-WSe$_2$ (black) placed on top of GaP nano-antennas with $r = 300$ and 100 nm, respectively. Spectra in red are measured on 1L- (**a**) and 2L-WSe$_2$ (**b**) placed on the planar GaP. Their intensity is multiplied by 10. **c** Experimental PL enhancement factor, $\langle \mathrm{EF} \rangle$, as a function of the antenna radius, for 1L- (orange) and 2L-WSe$_2$ (black). **d**–**g** Results of simulations showing various parameters determining the observed PL enhancement in a dimer as a function of the pillar radius (see main text and 'Methods' for further details). The values shown in the figures correspond to the behaviour of an oscillating electric dipole placed at the edge of the gap between the two pillars, 0.5 nm above the top of the pillar, and aligned along the line connecting the centres of the pillars [see inset in **e**]. **d** Enhancement of the excitation, $\gamma_{exc}/\gamma_{exc}^0$, dependent on the electric field intensity at the antenna surface and on the planar GaP substrate. **e** Enhancement of the radiative recombination rate, $\gamma_r/\gamma_r^0$. Inset shows variation of this ratio as a function of the dipole position above the dimer gap. The values plotted in **d**–**g** are calculated for the dipole placed at the position where $\gamma_r/\gamma_r^0$ reaches the maximum. **f** Enhancement of the light collection efficiency, $\eta/\eta^0$. **g** Dependence on $r$ of the calculated effective enhancement factor, $\langle \mathrm{EF} \rangle_{eff}$, defined as the product of the parameters in **d**–**f** (see Eq. (2) and the formula on the graph)

**Comparison of experimental data with the model.** In order to compare the results with our model (see 'Methods' and Supplementary Note IV), we introduce an effective enhancement factor $\langle \mathrm{EF} \rangle_{eff}$ defined as the product of three factors[15]:

$$\langle \mathrm{EF} \rangle_{eff} \propto \frac{\gamma_{exc}(\lambda_{exc})}{\gamma_{exc}^0(\lambda_{exc})} \cdot \frac{q(\lambda_{em})}{q^0(\lambda_{em})} \cdot \frac{\eta(\lambda_{em})}{\eta^0(\lambda_{em})} \qquad (2)$$

Here $\gamma_{exc}/\gamma_{exc}^0$ is the ratio of the excitation rates at a wavelength $\lambda_{exc}$, for an emitter coupled to the antenna ($\gamma_{exc}$) and placed on the planar substrate ($\gamma_{exc}^0$). Their ratio would account for the enhancement of the incident radiation leading to stronger light absorption in WSe$_2$. $\gamma_{exc} \propto (|E|/|E_0|)^2$, for which the spatial distribution is shown in Fig. 1b, c. We find that additional increase of the $\gamma_{exc}/\gamma_{exc}^0$ ratio arises from the reduction of $\gamma_{exc}^0$ for an emitter placed 0.5 nm above the planar GaP substrate compared with that for an emitter in the free space. The dependence of $\gamma_{exc}/\gamma_{exc}^0$ on the pillar radius $r$ is shown in Fig. 2d for $\gamma_{exc}$ calculated for an electric dipole placed 0.5 nm above the top surface of the pillars. The dipole is placed at the edge of one of the pillars just outside the gap. As shown in the inset in Fig. 2e and in Supplementary Note IV, this is the position where the coupling to the optical mode of the dimer is maximised. The data in Fig. 2d–f, showing the individual contributions of the different terms to the overall enhancement in Eq. (2), are calculated for this position of the dipole.

The second term describes the enhancement of the quantum yield ($q/q^0$) for an emitter at a wavelength $\lambda_{em}$. This is achieved through the enhanced rate of spontaneous emission[40] described by the Purcell factor $F_p = \gamma_r/\gamma_r^0$, where $\gamma_r$ and $\gamma_r^0$ are the rates of spontaneous emission for the emitter coupled to the antenna and placed on the planar GaP, respectively. In our model, we consider the limit of a low quantum yield for the emitter, i.e. the non-radiative decay $\gamma_{nr} \gg F_p \gamma_r$, which leads to $q/q^0 = F_p(\gamma_r + \gamma_{nr})/(F_p \gamma_r + \gamma_{nr}) \approx F_p$. The dependence on $r$ for this term is shown in Fig. 2e.

The third term $\eta/\eta^0$ describes the improved collection efficiency for WSe$_2$ PL on top of the nano-antennas ($\eta$) compared to planar GaP ($\eta^0$), as the emitted radiation is coupled to our detector using collection through numerical aperture NA = 0.7 (see Supplementary Note V). The dependence on $r$ of $\eta/\eta^0$ is shown in Fig. 2f.

Figure 2g shows the calculated values of the effective enhancement factor $\langle \mathrm{EF} \rangle_{eff}$ taking into account the above three mechanisms[15,40] (see 'Methods' and Supplementary Note IV). The dependences of $\langle \mathrm{EF} \rangle_{eff}$ and $\langle \mathrm{EF} \rangle$ are in a good qualitative agreement, suggesting that our model captures the main contributing factors.

**Polarisation-dependent luminescence.** We find further evidence for the sensitivity of the WSe$_2$ coupling to the optical mode of the

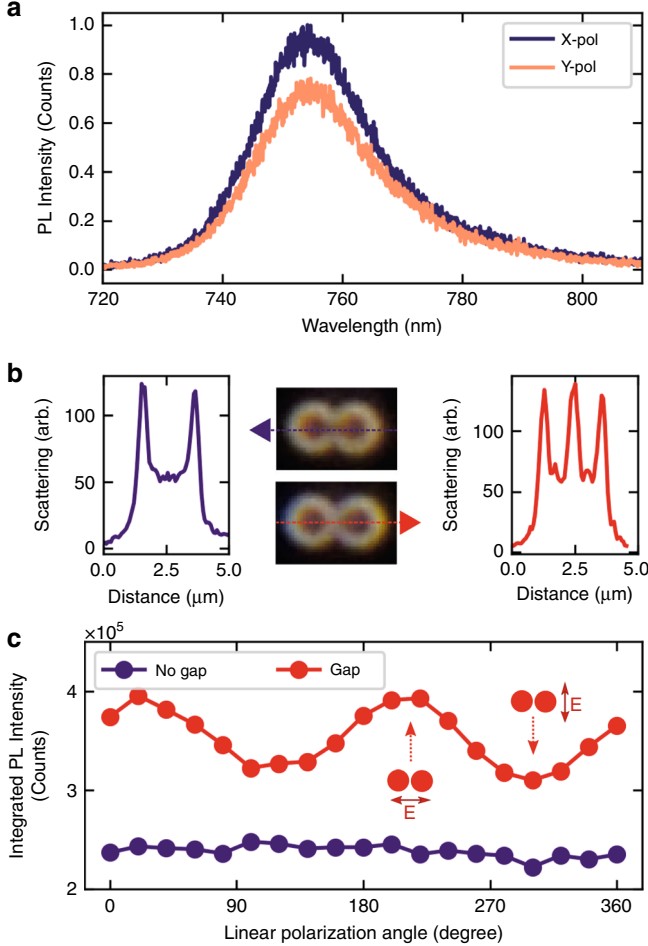

**Fig. 3** Polarisation properties of WSe$_2$ coupled to GaP nano-antennas. **a** Photoluminescence spectra for 1L-WSe$_2$ on a GaP nano-antenna ($g$ = 65 nm, $r$ = 400 nm) for a laser excitation linearly polarised along $X$- and $Y$-axes of the dimer (defined as in Fig. 1). **b** Dark-field microscopic images and related scattered light intensity profiles for two GaP nano-antennas ($r$ = 400 nm) covered with 1L-WSe$_2$. The scattered light intensity profiles are taken along the long axes ($Y$) of the dimer as shown by the dotted lines. The top (bottom) images and left (right) graphs correspond to the dimers without (with) a gap. **c** Comparison of 1L-WSe$_2$ PL intensity as a function of the orientation of linearly polarised excitation for the nano-antennas in **b**. Red (dark blue) shows the dependence for the dimer with (without) a gap

dimer in polarisation-resolved PL measurements. The spatial asymmetry of the dimer nano-antenna and the enhanced field in the gap between the two pillars is expected to lead to a polarisation-dependent response[32,37] as predicted by Fig. 1b, c. Such behaviour is found in PL in WSe$_2$ coupled to nano-antennas, as shown in Fig. 3a–c. Figure 3a shows the case for a nano-antenna with $r$ = 400 nm and a gap $g$ = 65 nm. It is observed that PL is 20% stronger when excited with an $X$-polarised laser compared to the $Y$-polarised excitation ($X$ and $Y$ are selected as in Fig. 1b, c). A similar modulation is observed in dimers with other values of $r$ (see Supplementary Note VI). The origin of this behaviour is further revealed when considering touching pillars with no gap. Figure 3b shows the dark-field microscopic images of dimers without (top) and with (bottom) a gap. The graphs in Fig. 3b show the extracted scattering intensities along the line connecting the centres of the pillars, revealing the absence (left) or the presence (right) of the gap. Figure 3c shows the integrated PL intensity for WSe$_2$ coupled to such dimers measured in a

micro-PL set-up. In the dimer with a gap, the PL is modulated by 20% when varying the polarisation of excitation, exhibiting higher intensity for $X$-polarised laser, which is due to the excitation of the optical mode in the gap between the pillars. A negligible polarisation dependence is observed in the dimer with no defined gap, emphasising nearly equal coupling of WSe$_2$ to the $X$- and $Y$-polarised optical modes.

**Radiative decay enhancement**. The theoretical prediction in Fig. 2 demonstrates that partly the PL enhancement originates from the enhanced radiative decay rate in WSe$_2$. We have been able to demonstrate this experimentally by measuring PL dynamics in WSe$_2$. Figure 4a shows the PL decay for a 1L-WSe$_2$ placed on the planar GaP (purple) compared to the emission when it is coupled to a nano-antenna with $r$ = 200 nm (orange). The curves are measured using excitation with a 90-ps pulsed laser at 638 nm and detection with an avalanche photo-diode detector (see 'Methods'). The corresponding instrument response function (IRF) is shown in Fig. 4a as a grey shaded area. For these measurements, we used a low excitation density (0.2 W/cm$^2$) to avoid non-radiative exciton–exciton annihilation, resulting in fast PL decay[2,42]. For low pumping powers, single exponential decay is usually observed, reflecting the radiative recombination dynamics of the thermalised exciton population[43,44]. Such behaviour is observed in Fig. 4a for the 1L-WSe$_2$ on the planar GaP, which shows a single exponential PL decay with a lifetime $\tau_{GaP} \approx 1.3$ ns. The PL decay measured for WSe$_2$ coupled to a dimer is dominated by a fast component with a lifetime $\tau_{dimer} \approx 0.2$ ns. This is very similar to the fast component in the IRF of $\tau_{IRF} \approx 0.16$ ns. We thus conclude that the measured $\tau_{dimer} \approx 0.2$ ns is a resolution-limited value. We interpret the shortening of the PL lifetime as a consequence of the Purcell enhancement of the radiative rate. A conservative lower-bound estimate of the Purcell enhancement factor is therefore ≈6. This is in a good agreement with the theoretical predictions, for which we should take into account the spatial variation of the Purcell enhancement in the WSe$_2$ coupled to the mode. Note that Fig. 2e only shows the values obtained for the optimum location of the dipole where the enhancement is maximised (see further details in Supplementary Note IV).

**Surface-enhanced Raman scattering**. Further evidence for the efficient interaction of WSe$_2$ with the nano-antennas is obtained from the observation of the enhanced Raman scattering response. This effect, typically observed for nano-structured metals[14] and dielectric nano-particles[45], is related to the surface localisation of the electromagnetic field. Both 1L and 2L-WSe$_2$ Raman spectra excited with a laser at 532 nm show a pronounced peak at 250 cm$^{-1}$ composed of two degenerate modes, the in-plane $E'$ for 1L and $E^1_{2g}$ for 2L WSe$_2$, and out-of-plane $A'_1$ for 1L and $A_{1g}$ for 2L WSe$_2$[46]. This is shown in Fig. 4b, c for 1L and 2L samples, respectively. Figure 4b, c further compare Raman spectra measured for WSe$_2$ on the planar GaP with the spectra collected on nano-antennas and show a notable enhancement of the signal. While the overall Raman signal decreases with decreasing $r$, its relative strength per unit nano-antenna area is strongly enhanced. In analogy with the PL data, we define the Raman experimental enhancement factor, $\langle EF \rangle^{Raman}$ (see 'Methods'). The obtained $\langle EF \rangle^{Raman}$ values increase when reducing the antenna radius, as for the PL enhancement, with values exceeding 10$^3$ for $r$ = 50 nm (Fig. 4d). The observed Raman intensity enhancement is explained by the enhancement of the laser and the scattered light fields and a more efficient collection of the signal (see Eq. (2)).

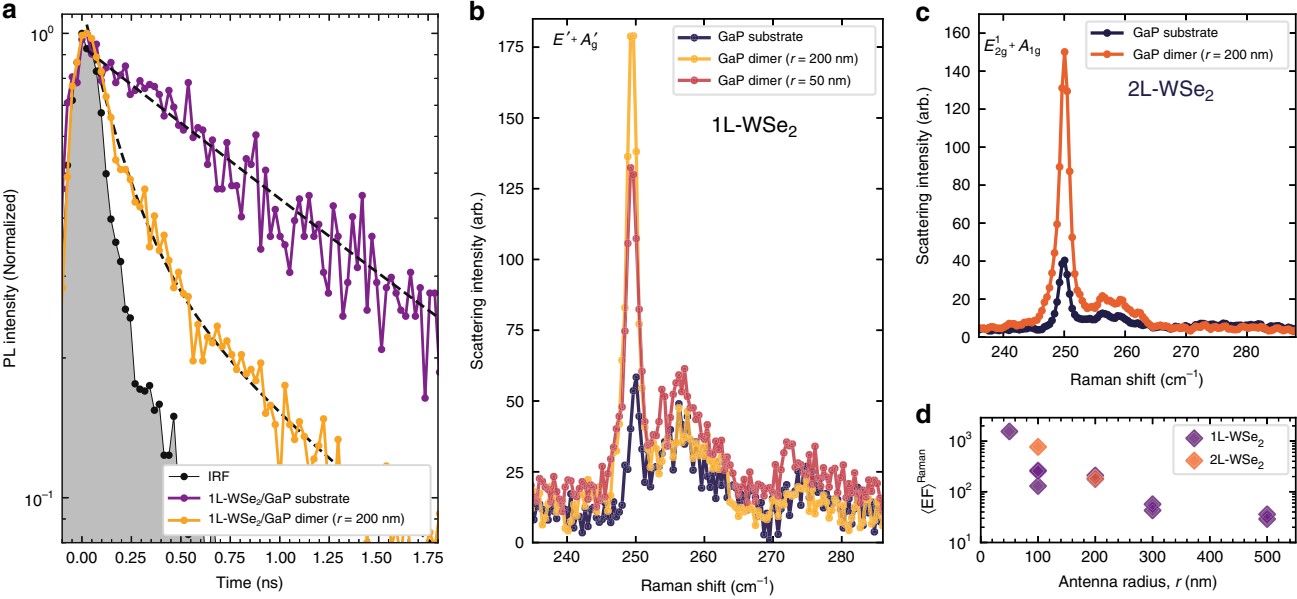

**Fig. 4** Time-resolved photoluminescence and Raman scattering spectra. **a** PL decay traces for a WSe$_2$ monolayer placed on a planar GaP substrate (purple) and coupled to a nano-antenna with $r = 200$ nm (yellow). Grey shows the instrument response function (IRF). The PL signal is measured with a 90-ps pulsed laser, under low excitation power of 0.2 W/cm$^2$, to avoid exciton–exciton annihilation processes. Resolution-limited dynamics are measured for WSe$_2$ coupled to a dimer. Dotted black lines show results of the fitting with exponential decay. **b** Raman scattering spectra for 1L-WSe$_2$ placed on GaP nano-antennas with $r = 50$ nm (red) and 200 nm (yellow) and on the planar GaP (black). **c** Raman scattering spectra for 2L-WSe$_2$ on a $r = 200$ nm GaP nano-antenna (orange) and on the planar GaP (black). **d** Experimental Raman enhancement factor, $\langle EF \rangle^{Raman}$ (see Eq. (3) in 'Methods'), obtained (after background subtraction) for the 250 cm$^{-1}$ peak in 1L- and 2L-WSe$_2$ coupled to nano-antennas with various $r$

## Discussion

The reported enhancement in the PL emission and Raman signal intensities and shortening of the radiative lifetime in 2D WSe$_2$ coupled to GaP nano-antennas show that nano-structured high-index dielectrics can be an efficient platform to engineer light–matter coupling on the nano-scale. Importantly, we show that the coupling to nano-antennas can be used to strongly enhance the quantum yield in TMDs via the enhancement of the radiative decay rate, emphasising the potential of this approach for light-emitting devices applications. The PL enhancements that we report are of the order or exceed those recently reported in TMDs coupled to metallic plasmonic nano-antennas, showing the viability of our approach in a broader nano-photonics context. Our approach could be further extended to arrays of dielectric nano-structures or meta-surfaces, an emerging field of nano-photonics[36], and to the use of van der Waals nano-photonics structures made from multilayer TMDs[35], with potential in a wide range of applications, such as quantum optics, photovoltaics, and imaging. The approaches for realisation of the Purcell effect in loss-less dielectrics demonstrated here will also be used in the field of strain-induced single photon emitters[47,48] for applications exploiting quantum light generation.

## Methods

**Sample fabrication**. GaP nano-antennas arrays are fabricated with the top–down lithographic process following the procedure in ref. [32]. The fabricated pillars are 200 nm high and have radii of $r = 50$, 100, 200, 300, 400, and 500 nm. Monolayers and bilayers of WSe$_2$ were mechanically exfoliated from a bulk crystal (HQGraphene) onto a poly-dimethylsiloxane polymer stamp. The thin layers are identified via the PL imaging technique described in this work in Fig. 1 and ref. [38]. The stamp was then used for an all-dry transfer of the exfoliated materials on top of the GaP nano-antenna array, in an home-built transfer set-up following ref. [49].

**PL imaging**. The PL imaging technique[38] used in Fig. 1 is obtained in a modified commercial bright-field microscope (LV150N Nikon). The white light source is used both for imaging the sample and as the PL excitation source. To excite the sample, a 550-nm short-pass filter is placed in the illumination beam path to

remove the near-IR part of the emission spectrum. The white light is then directed in a large numerical aperture objective (Nikon × 100 NA = 0.9) and collected with the same objective. The PL signal produced by the TMDs is isolated with a 600-nm long-pass filter before reaching the colour microscope camera (DS-Vi1, Nikon) used to image the sample.

**PL spectroscopy**. All PL spectra are acquired at room temperature in a micro-PL set-up with samples placed in vacuum. A 685-nm diode laser coupled to a single mode fibre is used as the excitation source. After passing through a 700-nm short-pass filter, the laser is focussed onto the sample through an infinity corrected objective (Mitutoyo × 100 NA = 0.7). For the data reported in Fig. 2, the average power entering the objective is 30 μW for monolayer PL and 120 μW for bilayer PL. The resulting laser spot has a radius of ≈3.5 μm, which is large enough to illuminate entire individual dimers, and kept constant for a uniform excitation of all the different nano-antennas sizes when determining the reported enhancement factor in Eq. (1). The dimers are separated by 10 μm, which allows optical measurement of an individual dimer. The emitted light is collected by the same objective and filtered with a 700-nm long-pass filter before coupling into a spectrometer (Princeton Instruments SP2750) and detection with a high-sensitivity liquid nitrogen cooled charge-coupled device (CCD; Princeton Instruments PyLoN). For polarisation measurements, a Glan–Thompson linear polarizer, followed by a half-wave plate mounted onto a motorised rotation stage, are inserted in the excitation path in order to control the linear polarisation angle of the laser source. No polarisation optics is used in the collection path.

**Time-resolved spectroscopy**. PL decay of planar WSe$_2$ is measured by coupling light filtered by a spectrometer (≈4 nm bandwidth) into a multi-mode fibre directing light to an avalanche photodiode (APD; ID100-MMF50) with time-resolution of ~40 ps. The signal from the APD is read using a photon counting card (SPC-130). An ≈90 ps pulsed diode laser (PicoQuant LDH) with wavelength 638 nm is used as the excitation source at a repetition rate of 80 MHz. Overall, the IRF has a full width at half maximum of <200 ps. The PL decay curves are fitted with single and bi-exponential decay functions of the form
$$y = y_0 + A_1 \exp^{-t/\tau_1} + A_2 \exp^{-t/\tau_2}.$$

**Raman spectroscopy**. Raman spectra are collected at room temperature in a micro-Raman set-up with samples placed in vacuum. A single mode 532-nm laser (Cobolt 04-01) is used, focussed on the sample through an infinity corrected objective (Mitutoyo × 50 NA = 0.55) with an average power of 200 μW before entering the objective. Background laser light is suppressed using three Optigate Bragg filters, which allow Raman signal to be measured by the same spectrometer/CCD system as for PL spectroscopy. The Raman signals are analysed by fitting the

two WSe$_2$ Raman peaks with Gaussian functions, following background subtraction. The 250 cm$^{-1}$ peak intensities extracted in this way are used to calculate the Raman enhancement factor in Eq. (3) given below. In this equation, $I_{Raman}^{fit}$ is the Raman intensity for WSe$_2$ coupled to a dimer and $I_{ref}^{fit}$ is for the reference WSe$_2$ on the planar GaP. We determined experimentally that $A_{laser} = 16.8\ \mu m^2$ in the micro-Raman set-up.

$$\langle EF \rangle^{Raman} = \frac{(I_{Raman}^{fit} - I_{ref}^{fit})}{A_{dimer}} \left( \frac{I_{ref}^{fit}}{A_{laser}} \right)^{-1} \tag{3}$$

**Simulations.** The effective enhancement factor contributions are calculated with a finite-difference time-domain (FDTD) method using the Lumerical FDTD solutions software. The near-field distribution is obtained by illuminating the structure with a plane wave, polarised along or perpendicular the dimer axis (the line connecting the centres of the pillars), and incident on the dimer perpendicular to the substrate from the vacuum side. The Purcell enhancement is calculated by exciting the structure with an oscillating electric dipole placed at different positions along the dimer axis at a vertical position 0.5 nm from the top surface of the GaP pillars and compared with a dipole placed 0.5 nm above the planar surface of the GaP substrate. To model the PL in WSe$_2$ originating from the in-plane excitons, the dipole is placed parallel to the substrate surface. The dipole oscillates along the line connecting the centres of the pillars in order to couple efficiently to the gap mode. The Purcell enhancement is calculated as the decay rate $\gamma$ of an emitter coupled to the GaP antenna, divided by the decay rate $\gamma_0$ of the same electric dipole emitter on the planar GaP substrate as described above. The decay rate enhancement $\gamma/\gamma_0$ corresponds to the enhancement of the rate of energy dissipation $P/P_0$. The collection efficiency is calculated by exciting the structure with an electric dipole, as above, and integrating the radiation pattern within the numerical aperture of the objective used in the experiments (NA = 0.7). The refractive index used for GaP is $n = 3.2$.

## Data availability
The data that support the findings of this study are available from the corresponding author upon reasonable request.

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

## Acknowledgements

L.S. and A.I.T. thank the financial support of the European Graphene Flagship Project under grant agreements 696656 and 785219, EPSRC grant EP/P026850/1 and the European Union's Horizon 2020 research and innovation programme under ITN Spin-NANO Marie Sklodowska-Curie grant agreement no. 676108. P.G.Z. and A.I.T. acknowledge support from the European Union's Horizon 2020 research and innovation programme under ITN 4PHOTON Marie Sklodowska-Curie grant agreement no. 721394. A.G. and A.I.T. acknowledge funding by EPSRC (EP/P026850/1). S.M., S.A.M. and R.S. acknowledge funding by EPSRC (EP/P033369 and EP/M013812). S.A.M. acknowledges the Lee-Lucas Chair in Physics and the DFG Cluster of Excellence Nanoscience Initiative Munich (NIM). M.A. and M.B. thank the DFG in the frame of TRR 142 for support.

## Author contributions

L.S. and P.G.Z. fabricated WSe$_2$ layers and transferred them on the GaP nano-antennas. L.S., P.G.Z. and A.G. carried out microscopy and CW optical measurements on WSe$_2$. L.S., D.S. and M.A. carried out time-resolved PL measurements. J.C. fabricated GaP nano-antennas. S.M., J.C., and R.S. carried out simulations for GaP nano-antennas. L.S., D.S., M.A., P.G.Z., A.G., M.B. and A.I.T. analysed optical spectroscopic data. M.B., S.A.M., R.S. and A.I.T. managed various aspects of the project. L.S. and A.I.T. wrote the manuscript with contributions from all co-authors. L.S., A.I.T., S.A.M. and R.S. conceived the idea of the experiment. A.I.T. oversaw the whole project.

## Competing interests

The authors declare no competing interests.
