## [Peer Review File · Nature Communications]

Reviewers' comments:

Reviewer #1 (Remarks to the Author):

Review for Nature Communications

Article title:

Enhanced light-matter interaction in an atomically thin semiconductor coupled with dielectric nano-antennas

Authors: L. Sortino et al.

This work is very interesting and potentially deserves publication in a top-level physical or interdisciplinary journal. The governing idea of placing the atomic monolayer (or a bilayer) of transition metal dichalcogenide on top of a dielectric (semiconductor) nanopillar seems to be novel and useful. The advantages granted by the electromagnetic coupling of a TMD sample with a dielectric (in the present paper GaP) nanoantenna of the suggested dual-pillar configuration compared to a usual flat substrate of a TMD and to plasmonic substrates are evident and properly discussed. The gain granted by such the NA to both photoluminescence and Raman radiation of the TMD is calculated and measured properly. This gain turned out to be higher than previously obtained for dielectric NAs. Authors obtained a noticeable radiative Purcell factor granted by such the NA to a quantum emitter. They have shown that their NA is strongly anisotropic -- its resonance holds when the electric field is polarized along the dimer axis. This anisotropy results in a partial polarization of the PL and polarization sensitivity for the Raman radiation proved experimentally.

However, this interesting and ambitious study is insufficient and the present version of the paper is unsuitable for publication in a so selective journal as Nature Comm. Though the technical solution is ingenious, and the authors have done a great job, the scientific level of this report is not very high. Additional studies are necessary so that to justify the publication of the work in Nature Comm.

First, the role of a NA keeps cryptic in the paper. It seems that the authors do not have a clear insight of its operation. They only state that its role is constructive and it gives an enhancement to both PL and Raman radiation of the TMD. They try to relate this enhancement with the radiative Purcell factor (PF) of their NA. However, this attempt is not convincing. Really, they plot the scattering cross section of the NA versus the disk radius r for a fixed gap g of their dimer in Supp. Mat. (Fig. 1) from which it is clear that the NA with $r=100$ nm is still resonant in the range of the PL from tungsten selenide, whereas the antennas with larger r lose the resonant properties. This observation is in line with Fig. 3c, where the profile of the PF at the frequency of maximal PL is depicted for $r=50-500$ nm. The smallest radius grants the largest PF (varying from 5 to 15). The authors claim that the physics of the huge PL enhancement is the Purcell effect. However, $PF=15$ cannot explain the enhancement of PL by 4 orders of magnitude and Raman radiation by 3 orders of magnitude! Moreover, in Fig. 4b we clearly see that the Raman gain granted by a NA with $r=200$ nm is higher than that granted by a NA with $r=50$ nm? Why is it so if the NA with $r=200$ nm is practically not resonant?

Second, the authors neglect the role of the local intensity enhancement. Just above formula (2) they name their explanation of the huge enhancement via the Purcell effect for in-plane excitons in the TMD "our theory". However, to be a theory the explanation is too inaccurate. Definitely, besides of the Purcell effect there is also the local enhancement of the source field granted by the NA. For the Raman gain, the enhancement of the intensity at the source frequency is simply equal to the Purcell factor at the same frequency (recently strictly proved for any linear reciprocal structure).

Therefore, if the Raman shift is small, the Raman gain is approximately equal to the square power of the Purcell factor. As to PL, the similar multiplicative concept of the gain is applicable, because the pumping by a 685 nm laser is used and the spectral line of the source radiation is within the resonance band of the NA. Thus, the gain should be close to the square power of the Purcell factor also for the PL from the TMD.

Third, the authors neglect the dimensional effect of the TMD sample. An exciton in a finite-area monolayer of atoms located on top of the NA is not a point dipole quantum emitter used in the calculations of the Purcell factor. The exciton could be treated a dipole if and only if the model ("our theory") properly explained the obtained experimental values. It does not explain, even if we take into account the square powers (see above). Thus, the authors have to study the effect of the area and shape of their TMD sample and to study the effect of the GaP NA in more details.

Let them know that their dielectric dimer has a set of resonances of different nature. The fundamental dipole resonance they exploit is the quasi-static LC resonance when the polarization of the GaP nanopillars results in their capacitive (C) coupling, corresponding to the electric field concentration in the antenna gap, whereas the magnetic flux responsible for L concentrates inside the disks. If they complement their color map on Fig. 1b and c by similar color maps for the magnetic field intensity, they will see it, for sure. Besides that LC-resonance (which really spreads with growing r), there are Mie resonances of the pillars in the visible and IR ranges. For a $h=200$ nm high pillar of GaP the values of its radius $r=200-500$ nm (i.e. nanodisk geometry) correspond to partially overlapping electric and magnetic Mie resonance. These resonances hybridize in the dimer of pillars and what the authors see in their Suppl. Mat. Fig. 1 for $r=200-500$ nm are the tails of the hybridized resonances.

Since the Purcell factor of a NA of GaP even together with the local field enhancement are not capable to explain the obtained gains, the theoretical model must take into account the patterning of the TMD mono- and bilayer. The patch of TMD whose shape, in general, repeats that of the NA can be resonant, as well. Who knows, perhaps the Purcell factor of the TMD patch on a flat GaP substrate may even exceed that of the NA?

To sum up: the authors have to study the eigenmodes of the NA, and give the evidence that the Purcell factor of their LC-resonance is really higher than that of the Mie resonances. Here, an excuse for exploiting namely the LC-resonance instead of the Mie resonances could be fabrication problems. Next, the authors have to study the resonant conditions for the exciton due to the TMD monolayer and bilayer patterning. The role of the local intensity enhancement in the explanation of the NA gain (both Raman and PL) must be present. I assume, that the observed gain can be presented as a product of 3 factors: the local intensity enhancement at the source frequency, the Purcell factor of the NA and the Purcell factor of the patterned sample (at the luminescence and at the Raman frequencies). If I am wrong, and the patterning of tungsten selenide does not result in the resonance, the authors have to find another convincing explanation. For example, it can be the so-called bound state in continuum (BIC). It is possible because, in fact, the object of the study is not a single NA covered by TMD but an array of such NAs. Perhaps, their wave interaction results in the BIC for the whole structure. In any case, without a convincing explanation of the underlying physics which is absent now the work hardly deserves publication in Nature Comm.

Recommendation: major revision.

Reviewer #2 (Remarks to the Author):

This article demonstrates and discuss the coupling of all-dielectric nanoantennas with TMDC materials. The authors claim that the true novelty of this work is the first demonstration of

coupling between the two systems.

However, directionality effects and PL Purcell enhancement have been already demonstrated in Mie resonator-TMD systems (<https://www.nature.com/articles/s41566-018-0155-y>). Also there have been previous demonstration of coupling between TMDC and Mie nanoresonators (<https://pubs.acs.org/doi/pdf/10.1021/acsami.7b17112> and <https://www.nature.com/articles/s41565-019-0442-x>).

For this reason, this article can be considered mostly as incremental and thus should not be accepted for publication in Nature Communications.

Additionally, the authors wrote and motivate the importance of their research with the sentence:

"However, high optical losses due to the absorption in the metal remain the main obstacle for further development of hybrid TMD-plasmonic devices [19]."

I do not understand what type of device would benefit of TMD-dielectric. In fact, i) There are several reports where TMD fluorescence is largely enhanced in TMD-plasmonic systems; ii) TMD are stable at moderately high temperatures typically achieved by plasmonic heating (100K increase), thus stability is not an issue here; iii) plasmonic systems present larger electric field enhancement and smaller mode volumes than all-dielectric systems. All in all, I believe there is no advantage in using all-dielectric devices to boost the performance of TMD material.

The claim of the authors should thus be discussed quantitatively, as it would be fundamental for the impact of this article. In short, why TMD-Mie is better than TMD-plasmonic?

I have several points that the authors should discuss prior to submission to other journals:

1) The authors use deliberately some sentences such as

very strong electric fields of the sub-wavelength confined modes

an ultra-small volume within the dimer gap

But the system they consider has none of these characteristics. In fact, the gap size is relatively large (60nm), and the electric field enhancement is quite modest (the intensity is increased only of 15 according to Figure 1b-c).

2) Did the authors use linearly polarized light for the PL measurements of Figure 1? If so, why the PL presents an almost homogeneous ring around the particle?

3) The authors never show, if I am correct, any experimentally measured scattering spectrum for the dimer, nor the simulated electric field map at the excitation frequency (I assume that they show in Figure 1c the field at the TMD emission wavelength).

4) The enhancement factor the authors obtain is quite arbitrary. In fact, they assume $A_{\text{dimer}} = 2 \cdot \pi \cdot r^2$. But if one does this, one is implicitly assuming that the emission is generated from all the antenna, which is clearly thus not subwavelength. In contrast, if the signal would be generated by the smaller gap region it should not strongly depend on the antenna radius (but this is in contrast with figure 2c).

5) The authors do not discuss why different dimers with the same size present fluorescent differences larger than a factor of 5.

6) The simulated excitation enhancement they predict is larger than 100, meaning that they would expect roughly a 100 increase in the electric field intensity at the excitation wavelength. More details should be provided together with a map of the field to support this statement

7) The authors claim a x15 times better collection efficiency because of the antenna. However, this comparison is unfair, as it is known that a large refractive index substrate favors emission into the substrate rather than on the top hemisphere. Also, there is no description on how this number has been calculated

8) In the PL image of figure S2 c the brightest spots appear not to be in the gap region. How do the authors explain this?

9) I believe that the authors should explain in more details the method they used in their work to facilitate the reproducibility of their results.

Dear Editor,

We would like to resubmit our manuscript by Sortino L. et al. 'Enhanced light-matter interaction in an atomically thin semiconductor coupled with dielectric nano-antennas' for further consideration in Nature Communications. We would be grateful if the manuscript could be sent to the reviewers.

We trust that we have addressed all concerns of the reviewers as described in detail below, where we provide a step by step response to the reviewers' comments.

In the resubmitted version of our manuscript we highlight in blue the parts that have been modified.

Reviewer #1 (Remarks to the Author):

This work is very interesting and potentially deserves publication in a top-level physical or interdisciplinary journal. The governing idea of placing the atomic monolayer (or a bilayer) of transition metal dichalcogenide on top of a dielectric (semiconductor) nanopillar seems to be novel and useful. The advantages granted by the electromagnetic coupling of a TMD sample with a dielectric (in the present paper GaP) nanoantenna of the suggested dual-pillar configuration compared to a usual flat substrate of a TMD and to plasmonic substrates are evident and properly discussed. The gain granted by such the NA to both photoluminescence and Raman radiation of the TMD is calculated and measured properly. This gain turned out to be higher than previously obtained for dielectric NAs. Authors obtained a noticeable radiative Purcell factor granted by such the NA to a quantum emitter. They have shown that their NA is strongly anisotropic -- its resonance holds when the electric field is polarized along the dimer axis. This anisotropy results in a partial polarization of the PL and polarization sensitivity for the Raman radiation proved experimentally.

However, this interesting and ambitious study is insufficient and the present version of the paper is unsuitable for publication in a so selective journal as Nature Comm. Though the technical solution is ingenious, and the authors have done a great job, the scientific level of this report is not very high. Additional studies are necessary so that to justify the publication of the work in Nature Comm.

Response: We thank the referee for their positive remarks on our work as well as the constructive critique. We provide detailed answers to their comments below.

First, the role of a NA keeps cryptic in the paper. It seems that the authors do not have a clear insight of its operation. They only state that its role is constructive and it gives an enhancement to both PL and Raman radiation of the TMD. They try to relate this enhancement with the radiative Purcell factor (PF) of their NA. However, this attempt is not convincing. Really, they plot the scattering cross section of the NA versus the disk radius r for a fixed gap g of their dimer in Supp. Mat. (Fig. 1) from which it is clear that the NA with $r=100$ nm is still resonant in the range of the PL from tungsten selenide, whereas the antennas with larger r lose the resonant properties. This observation is in line with Fig. 3c, where the profile of the PF at the frequency of maximal PL is depicted for $r=50-500$ nm. The smallest radius grants the largest PF (varying from 5 to 15). The authors claim that the physics of the huge PL enhancement is the Purcell effect. However, $PF=15$ cannot explain the enhancement of PL by 4 orders of magnitude and Raman radiation by 3 orders of magnitude!

Response: Indeed, as the referee points out the Purcell factor that we find in our simulations is not sufficient to explain the observed PL and Raman enhancement. Instead, as expressed in Eq.2 and shown in Fig.2d-f of the main text, we attribute the PL enhancement to the three main factors: (1) the radiative Purcell factor; (2) the enhancement of the optical excitation, proportional to the increased local field intensity at the nano-antenna surface; (3) the redirection of the light emitted by WSe_2 leading to improved collection efficiency. The product of these factors (Eq.2, Fig.2g) compares favourably with the experimental data in Fig.2c.

Changes: In order to emphasize the relation between the presented experiment and model, we add labels 'experiment' and 'model' in Fig.2c and g, respectively.

Moreover, in Fig. 4b we clearly see that the Raman gain granted by a NA with $r=200$ nm is higher than that granted by a NA with $r=50$ nm? Why it is so if the NA with $r=200$ nm is practically not resonant?

Response: The data shown in Fig.4b are the raw spectra before normalisation by the effective area of the antenna. The lower effective area for the pillars with $r=50$ nm leads to an overall smaller Raman signal. The normalisation is carried out according to Eq.3. The corresponding normalised data are shown in Fig.4d.

Second, the authors neglect role of the local intensity enhancement. Just above formula (2) they name their explanation of the huge enhancement via the Purcell effect for in-plane excitons in the TMD “our theory”. However, to be a theory the explanation is too inaccurate. Definitely, besides of the Purcell effect there is also the local enhancement of the source field granted by the NA.

Response: We agree with the reviewer that the local enhancement of the field at the emitter position is very important. This is already included in the local intensity enhancement which corresponds to term 1 in Eq.2.

Changes: In the new text, we now refer to our calculations as a ‘model’ rather than ‘theory’.

For the Raman gain, the enhancement of the intensity at the source frequency is simply equal to the Purcell factor at the same frequency (recently strictly proved for any linear reciprocal structure). Therefore, if the Raman shift is small, the Raman gain is approximately equal to the square power of the Purcell factor. As to PL, the similar multiplicative concept of the gain is applicable, because the pumping by a 685 nm laser is used and the spectral line of the source radiation is within the resonance band of the NA. Thus, the gain should be close to the square power of the Purcell factor also for the PL from the TMD.

Response: As we explain when discussing Eq.2 and Fig.2 in the manuscript, we consider 3 main contributions leading to the enhancement. The first one (Fig.2d) is the enhancement of the intensity of the excitation laser; second (Fig.2e), the Purcell enhancement for the emitter (TMD); third (Fig.2f), the effect of improved directionality (more light is emitted in the space above the substrate, where we collect light). As the reviewer states, the excitation enhancement and the Purcell enhancement lead to a multiplicative effect. Here instead of approximating it with the Purcell squared as suggested, we prefer to calculate the exact individual contributions (Purcell and excitation enhancement) and we multiply them, as in Eq 2.

Third, the authors neglect the dimensional effect of the TMD sample. An exciton in a finite-area monolayer of atoms located on top of the NA is not a point dipole quantum emitter used in the calculations of the Purcell factor. The exciton could be treated a dipole if and only if the model (“our theory”) properly explained the obtained experimental values. It does not explain, even if we take into account the square powers (see above). Thus, the authors have to study the effect of the area and shape of their TMD sample and to study the effect of the GaP NA in more details.

Response: Contrary to the referee’s statement, we trust that our model provides a good agreement with the experiment, as summarised in Fig.2. Both the order of magnitude of the effect and the dependence on the pillar radius have been reproduced well.

Regarding the modelling of the excitons as point dipoles, we believe this is a good approximation as the Bohr radius of the exciton in WSe₂ is around 1 nm, significantly smaller relative to any other dimensions of the system. An extended exciton, as well as exciton diffusion, may only reduce the observed enhancement due to spatial averaging.

We are not sure what the referee means by the ‘shape’ of our TMD sample. We comment below, with an additional calculation, on any photonic effect arising from a particular shape of the TMD layer. We believe these effects are negligible.

Let them know that their dielectric dimer has a set of resonances of different nature. The fundamental dipole resonance they exploit is the quasi-static LC resonance when the polarization of the GaP nanopillars results in their capacitive (C) coupling, corresponding to the electric field concentration in

the antenna gap, whereas the magnetic flux responsible for L concentrates inside the disks. If they complement their color map on Fig. 1b and c by similar color maps for the magnetic field intensity, they will see it, for sure. Besides that LC-resonance (which really spreads with growing r), there are Mie resonances of the pillars in the visible and IR ranges. For a $h=200$ nm high pillar of GaP the values of its radius $r=200-500$ nm (i.e. nanodisk geometry) correspond to partially overlapping electric and magnetic Mie resonance. These resonances hybridize in the dimer of pillars and what the authors see in their Suppl. Mat. Fig. 1 for $r=200-500$ nm are the tails of the hybridized resonances.

Response: We are very grateful to the referee for providing this information.

Changes: We now provide results of calculations for both the electric and magnetic fields in the Supplementary Note I.

Since the Purcell factor of a NA of GaP even together with the local field enhancement are not capable to explain the obtained gains, the theoretical model must take into account the patterning of the TMD mono- and bilayer. The patch of TMD whose shape, in general, repeats that of the NA can be resonant, as well. Who knows, perhaps the Purcell factor of the TMD patch on a flat GaP substrate may even exceed that of the NA?

Response: We trust that our model provides satisfactory explanation for the origin of the observed enhancement. This is demonstrated in Fig.2, where the results of our experiment (Fig.2c) and the model (Fig.2g) are presented. We agree that in principle there may be an optical effect due to the high index of the TMD. However, given that we deal with one and two atomic layer thick structures placed on a high index material (GaP), the expected effect is negligible. Below we show a plot of the FDTD calculated scattering cross section for a GaP dimer (blue) and a similar GaP dimer covered with a 1 nm thick layer of WSe₂ (dotted red), which show a very weak effect of the TMD within a few %.

To sum up: the authors have to study the eigenmodes of the NA, and give the evidence that the Purcell factor of their LC-resonance is really higher than that of the Mie resonances. Here, an excuse for exploiting namely the LC-resonance instead of the Mie resonances could be fabrication problems.

Response: Indeed, as the reviewer states, the dimer has both the gap and extended resonances originating from the mode hybridisation. We stress that both are accounted for in our model, which is a full solution of the Maxwell's equations. We have added a figure in the Supplementary Note I and

below to show the distribution of the electric and magnetic field on the dimer which as predicted by the reviewer is localised either in the gap (for E-field) or in the pillars (for H-field).

Next, the authors have to study the resonant conditions for the exciton due to the TMD monolayer and bilayer patterning.

Response: The monolayer has a negligible effect on the resonant conditions as shown in the scattering cross-section provided above. Exciton Bohr radius is 1 nm, and therefore it should generally be insensitive to the lateral size of the TMD layer considerably exceeding this value.

The role of the local intensity enhancement in the explanation of the NA gain (both Raman and PL) must be present. I assume, that the observed gain can be presented as a product of 3 factors: the local intensity enhancement at the source frequency, the Purcell factor of the NA and the Purcell factor of the patterned sample (at the luminescence and at the Raman frequencies). If I am wrong, and the patterning of tungsten selenide does not result in the resonance, the authors have to find another convincing explanation.

Response: We are worried that the referee has partially overlooked the interpretation of our results, which we presented in the manuscript. We explain in detail the factors that we have taken into account when presenting Eq.2 and Fig.2. As we explain when discussing Eq.2 and Fig.2 in the manuscript, we consider 3 main contributions leading to the enhancement. The first one (Fig.2d) is the enhancement of the intensity of the excitation laser; second (Fig.2e), the Purcell enhancement for the emitter (TMD); third (Fig.2f), the effect of improved directionality (more light is emitted in the space above the substrate, where we collect light). All three factors together provide good explanation for the behaviour which we see in the experiment. As shown above for the calculated scattering cross section for a GaP dimer with a 1 nm thick layer of WSe₂ (dotted red), the effect of WSe₂ is negligible. The effect of the coupling between different dimers (if this is what the referee means by the 'patterned sample') is also negligible, as they are positioned very far away from each other.

For example, it can be the so-called bound state in continuum (BIC). It is possible because, in fact, the object of the study is not a single NA covered by TMD but an array of such NAs. Perhaps, their wave interaction results in the BIC for the whole structure. In any case, without a convincing explanation of the underlying physics which is absent now the work hardly deserves publication in Nature Comm.

Response: We can rule out BIC due to the large separation between individual dimers (10 micron). We see no evidence for occurrence of any such effects. The enhancement that we observed is explained well by our model without a need for taking into account any other effects.

Reviewer #2 (Remarks to the Author):

This article demonstrates and discuss the coupling of all-dielectric nanoantennas with TMDC materials. The authors claim that the true novelty of this work is the first demonstration of coupling between the two systems.

However, directionality effects and PL Purcell enhancement have been already demonstrated in Mie resonator-TMD systems (<https://www.nature.com/articles/s41566-018-0155-y>). Also there have been previous demonstration of coupling between TMDC and Mie nanoresonators (<https://pubs.acs.org/doi/pdf/10.1021/acsami.7b17112> and <https://www.nature.com/articles/s41565-019-0442-x>).

For this reason, this article can be considered mostly as incremental and thus should not be accepted for publication in Nature Communications.

Response: We thank the referee for their comments and the constructive critique of our work, which resulted in a more balanced and informative manuscript.

We are grateful for bringing the papers mentioned above to our attention. We argue that our work is not just an incremental step in the new research field exploring the coupling of TMDs to various photonic structures. Instead our findings open a new important direction in this field. Our main result concerns large enhancement of PL and Raman signal in mono- and two-layer TMD coupled to nano-antennas made of a high index and large band-gap material (GaP). As we explain below, this is a considerably different direction to the results published in these three papers.

- In the work by Cihan et al only the effect of the modified directionality of the emission from MoS₂ coupled to a Si nanowire has been reported. No PL enhancement and Purcell effect has been experimentally observed. Light emitted by MoS₂ will in principle be partially absorbed in Si. Use of Si structures for this particular TMD family is undesirable if PL enhancement is sought. We instead use nano-antennas made from GaP, which previously have not been used for light-matter interaction experiments in 2D materials.

- Lepeshov et al report mostly on theoretical investigations of magnetic Mie resonances in Si nanoparticles. Very limited experimental evidence is presented. No PL enhancement or Purcell effect in the TMD was reported.

- The work by Verre et al is not related to the coupling of two dimensional TMDs to nano-antennas. Here a bulk TMD is used as a medium from which the nano-pillars are made, and the field is strongly enhanced inside the nano-pillar. The coupling with light is reported for excitons in a thick multi-layer TMD. These bulk excitons have considerably different properties from the 2D excitons, which we study, and whose properties are strongly modified by the confinement and (important for our work) strain. No PL enhancement or Purcell effect in the TMD was reported by Verre et al.

Changes: In the new version of our manuscript, we cite all three papers and modify the introduction accordingly in the place where we discuss high-refractive-index dielectric nano-antennas.

Additionally, the authors wrote and motivate the importance of their research with the sentence:

"However, high optical losses due to the absorption in the metal remain the main obstacle for further development of hybrid TMD-plasmonic devices [19]."

I do not understand what type of device would benefit of TMD-dielectric. In fact, i) There are several reports where TMD fluorescence is largely enhanced in TMD-plasmonic systems; ii) TMD are stable at moderately high temperatures typically achieved by plasmonic heating (100K increase), thus stability is not an issue here; iii) plasmonic system present larger electric field enhancement and smaller mode volumes than all-dielectric systems. All in all, I believe there is no advantage in using all-dielectric devices to boost the performance of TMD material. The claim of the authors should thus be discussed quantitatively, as it would be fundamental for the impact of this article. In short, why TMD-Mie is better than TMD-plasmonic?

Response: The strongest photoluminescence enhancements which we report exceeds 10000 (30000 for 2L structures). This is comparable with the highest reported for TMDs coupled to plasmonic structures. For example, Wang et al Nature Communications 7, 11283 (2016) report an enhancement of 20000. Note, that the very large enhancements reported by Wang et al are calculated relative to WSe₂ placed on gold, which is known to lead to strong quenching of PL in TMDs.

Other papers, for example Kern et al ACS Photonics 2, 1260-1265 (2015), Lee et al Nano Lett. 15, 3646-3653 (2015), Butun et al Nano Lett.15, 2700-2704 (2015) etc, usually report ‘as-measured’ PL enhancement (from 2 to 10 times) without normalising it to the size of the nanoparticle or its plasmonic mode. In our case such enhancement factors also exceed 10 and reach up to 50 as shown in Fig.2a,b. It can also be estimated that if recalculated to take into account the size of the mode, these papers achieve similar enhancements to Wang et al.

In studies with plasmonic structures, it is also a common approach to use functionalized metallic nano-particles, where the metal and the TMD layer are separated by a spacer suppressing PL quenching [see e.g. Kern et al ACS Photonics 2, 1260-1265 (2015)] or a dielectric spacer serving the same purpose is deposited between the metal and the TMD [see e.g. Cheng et al ACS Photonics 4, 1421-1430 (2017), Cai et al ACS Photonics 5, 3466-3471 (2018), Luo et al Nature Nanotechnology 13, 1137–1142 (2018), Tahersima et al ACS Photonics 4, 1713-1721 (2017), etc].

Our work shows that the use of dielectric antennas opens a very promising new avenue for *light emitting* applications, which can surpass plasmonics due to negligible detrimental proximity effects including (i) negligible quenching, especially important for quantum applications, (ii) minimal unwanted effects (including chemical effects) due to free-electrons and hot-electrons.

We emphasize, that the low losses in all-dielectric cavities is a very attractive feature. Now, that we have demonstrated similar PL enhancement to plasmonic nano-antennas, there is an additional motivation to study them further and improve their performance, for example with ultra-narrow gaps or using inverse-designed structures (Mignuzzi et al. *Nano Lett.* 19 1613 (2019)).

Further possibilities in a long-term may be open by potentially CMOS compatible fabrication of all-dielectric nano-antennas.

The structures based on nano-antennas proposed and studied here are very well suited for several light emitting applications such as for example electrically pumped light emitting TMD/graphene/hBN heterostructures [see e.g. our previous work in Nature Materials 14, 301 (2015) and Nano Letters 15, 8223 (2015)] as well as strain-activated single photon emitters as was recently reported by Cambridge and Edinburgh groups [Nature Communications 8, 15093 and 15053 (2017)]. Thus, our findings open a new direction for various *light emitting* devices made from 2D materials.

Changes: We have removed the phrase that the referee mentions above. We have now provided a much fuller account of the experimental work where TMDs were coupled to plasmonic nano-structures in order to provide a broader context for our study. The main corresponding changes in the text are in the abstract and introduction. We comment in the conclusions that the PL enhancements, which we report are comparable or higher than those obtained using plasmonic nano-antennas.

I have several points that the authors should discuss prior to submission to other journals:

1) The authors use deliberately some sentences such as
very strong electric fields of the sub-wavelength confined modes
an ultra-small volume within the dimer gap

But the system they consider has none of these characteristics. In fact, the gap size is relatively large (60nm), and the electric field enhancement is quite modest (the intensity is increased only of 15 according to Figure 1b-c).

Response: The description we used was prompted by comparison with the well-studied dielectric structures, such as Fabry-Perot cavities, photonic crystals etc, where the photonic modes have at best diffraction limited sizes $\sim \lambda/n$. In comparison to those structures, the description which we used was appropriate. The referee’s remark is probably based on comparison of our nano-antennas to typical plasmonic structures. In that regard we agree that the description we used may be less appropriate.

Changes: We removed or softened all such descriptions in the text.

2) Did the authors used linearly polarized light for the PL measurements of Figure 1? If so, why the PL presents an almost homogeneous ring around the particle?

Response: In Figure 1 the PL imaging is achieved by using an unpolarised white light source in a commercial optical microscope. A detailed description of the imaging method is given in the Methods section and in our earlier work: Alexeev et al. Nano Lett. 17, 5342 (2017), Ref.[24] in the main text.

Changes: We have updated the description of Fig.1 in the revised manuscript to explain better that unpolarised light was used.

3) The authors never show, if I am correct, any experimentally measured scattering spectrum for the dimer, nor the simulated electric field map at the excitation frequency (I assume that they show in Figure 1c the field at the TMD emission wavelength).

Response: Experimental scattering spectrum of similar devices fabricated by the same authors have been reported in our previous work by Cambiasso et al. Nano Lett. 17, 1219 (2017), Ref. [34] in the current manuscript version. In Figs 1b,c we show the field at the excitation wavelength of 685 nm.

Changes: We have changed Fig.1b,c and updated its description in text and in the caption to explain better what is shown in the figure.

4) The Enhancement factor the authors obtain is quite arbitrary. In fact, they assume $A_{dimer} = 2 \cdot \pi \cdot r^2$. But if one does this, one is implicitly assuming that the emission is generated from all the antenna, which is clearly thus not subwavelength. In contrast, if the signal would be generated by the smaller gap region it should not strongly depend on the antenna radius (but this is in contrast with figure 2c).

Response: We find that the enhanced PL comes not only from the gap region, but also from the edges of the pillars, as shown for example in Fig.1f, and could be expected based on the field distribution shown in Figs.1b,c. Thus, using the normalisation procedure which accounts for different radii of the antennas allows us to provide a direct comparison between structures of different sizes. We agree with the referee that a choice of the effective area over which normalisation should be made is non-trivial and somewhat arbitrary. Using $A_{dimer} = 2\pi r^2$ for normalisation is one of the approaches, which provides a lower bound for the enhancement factor. Surprisingly, values of A_{dimer} calculated in this way correspond closely to the ones obtained from a more elaborate calculation given in Ref.[40]. Even for the largest pillar $A_{dimer} \approx (\lambda_{exc})^2$, whereas for $r \approx 100$ nm $A_{eff} \ll (\lambda_{exc})^2$.

5) The authors do not discuss why different dimers with the same size present fluorescent differences larger than a factor of 5.

Response: The difference in the fluorescence intensity can be related to the factors listed below.

(1) Non-uniformity of the coupling between WSe₂ and the nano-antennas. This may be caused by a variety of factors such as local contamination from the polymer used for the WSe₂ transfer, local deformation of the WSe₂, local presence of water etc.

(2) Non-uniformity of the structural properties of the nano-antennas. For example, the size of the gap may vary. The quality of etching may also vary, for example producing sidewalls of the pillars, which are not perfectly vertical etc.

Changes: In the original version of the manuscript we provided only a very brief description of this observation. We have now expanded this description.

6) The simulated excitation enhancement they predict is larger than 100, meaning that they would expect roughly a 100 increase in the electric field intensity at the excitation wavelength. More details should be provided together with a map of the field to support this statement

Response: The enhancement factor shown in Fig.2d of the main text is calculated as a ratio $\gamma_{exc}/\gamma_{exc}^0$ (as explained in text), which corresponds to the ratio of the intensity enhancements on the nano-antenna and the planar GaP. The typical maps and values for the electric field enhancements around the nano-antennas can be inferred from Fig.1b,c, where $(|E|/|E_0|)^2$ is shown, with E being the electric field amplitude in the wave scattered by the pillars, and E_0 is the electric field amplitude of the normally incident wave. γ_{exc} follows a similar spatial distribution as it is proportional to $(|E|/|E_0|)^2$. However, the excitation rate γ_{exc}^0 for an emitter placed 0.5 nm above the planar GaP substrate turns

out to be reduced compared with that for an emitter in the free space. This leads to the ratio $\gamma_{exc}/\gamma_{exc}^0$ larger than the ratio $(|E|/|E_0|)^2$ in Fig.1b,c.

Changes: We updated the description of Fig.2 to expand the explanation of the meaning and the value of $\gamma_{exc}/\gamma_{exc}^0$.

7) The authors claim a x15 times better collection efficiency because of the antenna. However, this comparison is unfair, as it is known that a large refractive index substrate favors emission into the substrate rather than on the top hemisphere. Also, there is no description on how this number has been calculated.

Response: In our view, the comparison between the planar GaP substrate and the nano-antennas made of the same material is reasonable for the description of our experimental findings. Indeed, the referee is correct that an emitter placed on a high-index planar substrate would emit mostly into the substrate, as shown in the figures below (now also reported in Supplementary Information in the revised version of the manuscript). Our simulations show that the emission in the space above the substrate is considerably increased by the presence of the nano-antenna, which is a non-trivial result. Calculations for the collection efficiency are described in the Methods section.

Changes: We introduced a new section in the Supplementary Information showing an example of calculated emission patterns for emitters placed on a planar GaP substrate and near a nano-antenna.

8) In the PL image of figure S2 c the brightest spots appear not to be in the gap region. How do the authors explain this?

Response: The lower intensity in the gap region corresponds to the nano-antennas where due to fabrication imperfections the gaps have not fully opened. The partial loss of PL or Raman intensity leads to the scatter in the data in Fig.2c that the referee noticed in their question 5 answered above.

9) I believe that the authors should explain in more details the method they used in their work to facilitate the reproducibility of their results.

Response: In this paper we report results obtained on two substrates, where we measured 30 dimers of the same double-pillar geometry and similar sizes (Fig.2c present results for 26 dimers measured in the same experimental set-up and the same conditions). We have conducted many more experiments on similar samples where we studied different effects (such as low temperature behaviour etc), but all samples showed similar behaviour to the one reported here. The results are consistent, with deviations explained by fabrication imperfections. We have not encountered any issues with the reproducibility of the results, except for those that the referee asked about in question 5 above.

REVIEWERS' COMMENTS:

Reviewer #1 (Remarks to the Author):

Since the initial manuscript was a report of physical effects in an interesting novel structure I was positive about its possible publication in Nature Comm. I fully disagree with the 2d referee who thinks that the novelty of this paper is incremental. The Purcell effect was predicted and experimentally confirmed in 1940s and since that time hundreds of great papers were published about unusual features of this basic effect in unusual structures. The authors cover a previously known dielectric dimer nanoantenna with a 2D material and earn qualitatively new properties which result in exciting features of the Purcell effect, theoretically and experimentally studied for a single-molecule fluorescence. I think, it is a prerequisite for publishing the report of the study in a top-class physical or even interdisciplinary journal.

The authors have done a good job revising the paper. They have managed to persuade me that their explanations are correct. The comments of the 2d reviewer were also instructive and the paper in general became much better. I would recommend its publication in its present form.

Reviewer #2 (Remarks to the Author):

The authors spent a lot of energy for a careful and detailed reply on the points raised. I am happy and I believe that the article can now be published.